# Integrative *in vivo* analysis of the ethanolamine utilization bacterial microcompartment in *Escherichia coli*

Denis Jallet,[1] Vanessa Soldan,[2] Ramteen Shayan,[2] Alexandre Stella,[3,4] Nour Ismail,[1] Rania Zenati,[1] Edern Cahoreau,[1,5] Odile Burlet-Schiltz,[3,4] Stéphanie Balor,[2] Pierre Millard,[1,5] Stéphanie Heux[1]

**ABSTRACT** Bacterial microcompartments (BMCs) are self-assembling protein mega-complexes that encapsulate metabolic pathways. Although approximately 20% of sequenced bacterial genomes contain operons encoding putative BMCs, few have been thoroughly characterized, nor any in the most studied *Escherichia coli* strains. We used an interdisciplinary approach to gain deep molecular and functional insights into the ethanolamine utilization (Eut) BMC system encoded by the *eut* operon in *E. coli* K-12. The *eut* genotype was linked with the ethanolamine utilization phenotype using deletion and overexpression mutants. The subcellular dynamics and morphology of the *E. coli* Eut BMCs were characterized *in cellula* by fluorescence microscopy and electron (cryo)microscopy. The minimal proteome reorganization required for ethanolamine utilization and the *in vivo* stoichiometric composition of the Eut BMC were determined by quantitative proteomics. Finally, the first flux map connecting the Eut BMC with central metabolism *in cellula* was obtained by genome-scale modeling and $^{13}$C-fluxomics. Our results reveal that contrary to previous suggestions, ethanolamine serves both as a nitrogen and a carbon source in *E. coli* K-12, while also contributing to significant metabolic overflow. Overall, this study provides a quantitative molecular and functional understanding of the BMCs involved in ethanolamine assimilation by *E. coli*.

**IMPORTANCE** The properties of bacterial microcompartments make them an ideal tool for building orthogonal network structures with minimal interactions with native metabolic and regulatory networks. However, this requires an understanding of how BMCs work natively. In this study, we combined genetic manipulation, multi-omics, modeling, and microscopy to address this issue for Eut BMCs. We show that the Eut BMC in *Escherichia coli* turns ethanolamine into usable carbon and nitrogen substrates to sustain growth. These results improve our understanding of compartmentalization in a widely used bacterial chassis.

**KEYWORDS** *Escherichia*, microcompartment, ethanolamine, catabolism, *eut* operon

ompartmentalization was initially considered a defining feature of eukaryotic cells. With the advent of transmission electron microscopy (TEM) however, subcellular structures were also discovered in bacteria, including bacterial microcompartments (BMCs) such as carboxysomes in certain autotrophic taxa (1). Genes encoding for BMC shell proteins were subsequently discovered in heterotrophic taxa (2, 3) and recent isolate and metagenomics data suggest that roughly 20% of bacteria have BMC-encoding operons (4). These genes are generally found in large operons with the other genetic components required to produce BMCs, often referred to as metabolosomes, because of their typically catabolic rather than anabolic function. The ethanolamine utilization (Eut) BMC in *Salmonella enterica* was one of the first studied metabolosome systems (5).

**Editor** Ákos T. Kovács, Universiteit Leiden, Leiden, the Netherlands

Address correspondence to Denis Jallet, denis.jallet@insa-toulouse.fr.

The authors declare no conflict of interest.

See the funding table on p. 21.

Ethanolamine (EA) accumulates in mammalians' gastrointestinal and urinary tracts upon degradation of phosphatidylethanolamine. EA utilization confers competitive advantages to certain pathogenic bacteria in the gut environment (6–8). Some *Escherichia coli* strains can utilize EA as the sole nitrogen and carbon source in the presence of vitamin B12 (6, 9–12). EA utilization was initially thought to be associated with pathogenicity (6, 13) but several commensal *E. coli* strains also metabolize EA (11, 12). The *E. coli* core genome natively hosts a 17-gene *eut* operon (*eutSPQTDMNEJ-GHABCLKR*) with strong homology to the *eut* operon of *S. enterica* (14, 15). Other genes, such as *maeB, talA, or tktB*, flank the *eut* operon to form an extended EUT1 locus (16, 17). The EUT1 locus is well conserved amongst various Beta- and Gamma-proteobacteria (4), suggesting a functional link between at least some of these ancillary EUT1 proteins and EA utilization (16). However, the link has not been evaluated experimentally yet. EUT loci with different *eut* operon arrangements and ancillary gene contents also exist amongst bacteria (i.e., EUT2 and EUT3 loci [4, 16]). Since *E. coli* only has an EUT1 locus, EUT1 will be the focus of the present study.

The *eut* operon contains genes encoding for BMC components. The Eut BMC shell is made up of ring-shaped oligomeric proteins (EutSMNLK), some of which have central pores that allow some small metabolites to pass, including EA and its degradation products (18). Inside Eut BMCs, EA is thought to be deaminated into acetaldehyde plus ammonium ($NH_4$) by ethanolamine ammonia-lyase (EAL: EutBC), but the exact metabolic topology involved remains debated (19, 20). Acetaldehyde is further converted through enzymatic reactions. $NH_4$, ethanol, and acetyl phosphate (acetyl-P) would eventually diffuse out through the shell to be used in central metabolism or excreted (19). But again, the exact BMC core protein complement remains hypothetical. Two Eut enzymes (EutC, EutE) bear encapsulation peptides that likely favor their internalization (21, 22). Other Eut enzymes may form complexes or interact with shell components.

Electron micrographs of *E. coli* Eut BMCs have only ever been reported twice (9, 23). Indeed, while irregular polyhedral structures with a diameter of around 100 nm can be observed in *E. coli* K-12 and *E. coli* UPEC U1 after *eut* induction, it is unclear whether these Eut BMCs are well formed and functionally required for EA catabolism. Intact Eut BMCs have never been purified to date, either from *E. coli* or from *S. enterica*, and our knowledge of these systems mainly comes from genetic studies in *S. enterica* (19, 20, 24) and *in vitro* structural and functional characterizations of individual recombinant Eut proteins (18).

Overall, Eut BMCs play a safeguarding role by sequestering and favoring the downstream conversion of toxic and volatile acetaldehyde intermediates (20). Their spatial arrangement favors metabolic channeling and it has been reported that compartmentalization may increase catalytic activity by up to sixfold compared with non-compartmentalized reactions (25). Another advantage is that the cofactors required for enzyme activity are regenerated within the BMC thus limiting competition with enzymes from the cytosol (19). These properties make Eut BMCs and BMCs interesting metabolic units for biotechnological applications (26), but a better understanding of their regulation, assembly, and function is still required to make them amenable to engineering.

With these knowledge gaps in mind, we performed a systemic analysis from the molecular to the functional level of native BMC-mediated EA catabolism in *E. coli* K-12. We characterized wild-type (WT) *E. coli* K-12 W3110 and several mutant strains by epifluorescence microscopy, TEM, and electron cryotomography (cryoET) and performed a multi-omics analysis to understand how BMC-mediated EA utilization affects bacterial physiology.

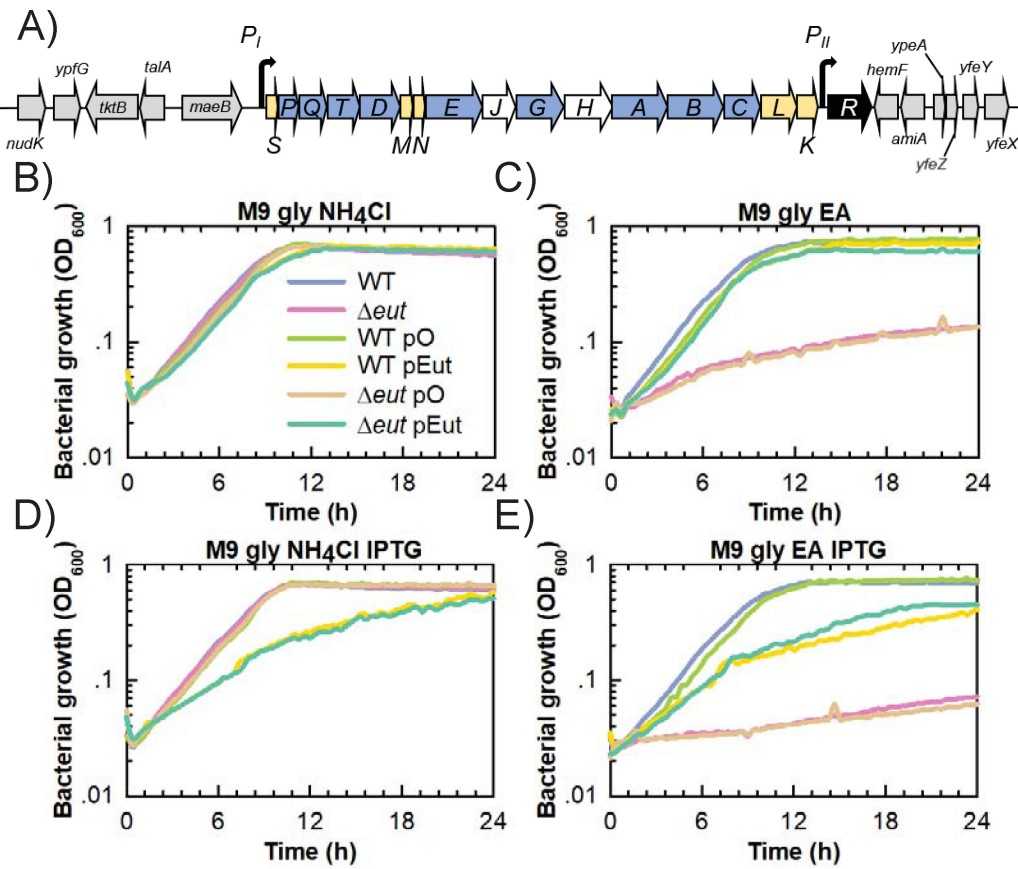

**FIG 1** The *eut* operon is essential for EA utilization as a nitrogen source by *E. coli* K-12 W3110. (A) Schematic representation of the chromosomal EUT1 locus in W3110 WT. Genes in gray flank the *eut* operon and are conserved amongst the EUT1 loci of Beta- and Gammaproteobacteria. Genes with other colors form the *eut* operon. Genes colored in yellow encode structural components of the Eut BMC shell, those in blue encode for enzymes (EA catabolism, coenzyme B12 biosynthesis), and those in white encode other functions (chaperon, transporter). (B–E) Plate reader measurements of optical density at 600 nm for strains grown in (B) M9 medium containing glycerol (30 mM) as C source and $NH_4Cl$ (20 mM) as N source; (C) M9 medium containing glycerol (30 mM) as C source and EA (20 mM) as N source, including vitamin B12 (200 nM); (D) M9 glycerol $NH_4Cl$ with IPTG (20 µM) and (E) M9 glycerol EA B12 with IPTG (20 µM). All the strains and plasmids are described in Table 1. Gentamicin (10 µg·mL$^{-1}$) was added for the plasmid-bearing strains. The data shown are from a representative experiment of at least three independent replicates.

## RESULTS

### Necessity of the *eut* operon for EA utilization

The EUT1 locus of *E. coli* K-12 W3110 (Fig. 1A) is similar in gene content to *S. enterica* LT2's, with more than 73% primary structure identity between individual EUT1 proteins from the two species (Table S2). In contrast, the *eut* operon of *E. coli* K-12 BW25113 is natively interrupted by a prophage DNA insertion downstream of *eutA* (Fig. S1A). *E. coli* K-12 does not contain any other putative BMC locus besides EUT1 (e.g. no PDU locus). To investigate the link between *eut* genotype and EA catabolism, we first constructed several mutant strains by genome editing (27). The *eut* operon was knocked out from WT W3110 to yield W3110 Δ*eut* (Table 1; Fig. S1B). The prophage DNA insertion in BW25113 WT was scarlessly removed to reconstitute an intact *eut* operon, thereby generating BW25113 *eut+*. We additionally built *eut* overexpressors by cloning the entire W3110 WT *eut* operon downstream of an isopropyl-β-D-thiogalactoside (IPTG)-inducible promoter [P$_{trc}$ (28)] in pSEVA661 (29), yielding pEut (Fig. S1C). pEut was transformed into W3110 Δ*eut* for complementation assays.

**TABLE 1** Bacterial strains and plasmids used in the study

| Strain or plasmid | Genotype or description | Source |
|---|---|---|
| *E. coli* bacterial strains | | |
| W3110 WT | *E. coli* K-12 W3110 WT [F⁻ λ⁻ INV(*rrnD-rrnE*) *rph-1 eut*⁺] | Laboratory collection |
| BW25113 WT | *E. coli* K-12 BW25113 WT [F⁻ λ⁻ Δ*lacZ4787*(::*rrnB-3*) *hsdR514* Δ(*araBAD*)*567* Δ(*rhaBAD*)*568 rph-1 eut*⁻(::cryptic prophage DNA)] | Laboratory collection |
| W3110 Δ*eut* | Chromosomal *eut* operon removed by genome editing in W3110 WT | This study |
| BW25113 *eut*+ | Cryptic prophage DNA insertion within the chromosomal *eut* operon removed by genome editing in BW25113 WT | This study |
| W3110 *eutC-GFP* | $P_{LtetO-1}$::*eutC-GFP* construct inserted at the SS9 safe chromosomal site in W3110 WT by genome editing | This study |
| W3110 *eutC$_{1-20}$-GFP* | $P_{LtetO-1}$::*eutC$_{1-20}$-GFP* construct inserted at the SS9 safe chromosomal site in W3110 WT by genome editing | This study |
| Plasmids | | |
| pCas | *repA101*(Ts), Kan$^R$, $P_{cas}$::*cas9* $P_{araB}$::*Red laclq* $P_{trc}$::*sgRNA-pMB1* | (22) |
| pTargetF | *pMB1*, Sp$^R$, *sgRNA-acs*; expression of a specific sgRNA to target *acs* in *E. coli* | (22) |
| pTargetF_eut | *pMB1*, Sp$^R$, *sgRNA-eutG*; pTargetF derivative for the expression of a specific sgRNA to target *eutG* in *E. coli* | This study |
| pHD_Δ*eut* | *pUC*, Amp$^R$, contains the 1-kb-long upper and lower homology arms required to knock out the chromosomal *eut* operon in W3110 WT | This study |
| pTargetF_SS9 | *pMB1*, Sp$^R$, *sgRNA-SS9*; pTargetF derivative for the expression of a specific sgRNA to target the SS9 locus in *E. coli* | This study |
| pHD_SS9_eutC-GFP | *pUC*, Amp$^R$, contains 1 kb-long homology regions to insert the $P_{LtetO-1}$::*eutC-GFP* construct at the SS9 safe chromosomal site in W3110 WT | This study |
| pHD_SS9_eutC$_{1-20}$-GFP | *pUC*, Amp$^R$, contains 1 kb-long homology regions to insert the $P_{LtetO-1}$::*eutC$_{1-20}$-GFP* construct at the SS9 safe chromosomal site in W3110 WT | This study |
| pSEVA661 | *p15A*, Gm$^R$ | (24) |
| pO | *laclQ*-$P_{trc}$ cassette and $T_0$ terminator cloned into the multiple cloning site of pSEVA 661 | This study |
| pEut | Entire *eut* operon from W3110 WT cloned into pO under the control of $P_{trc}$ | This study |

Growth experiments were performed using a microplate reader to evaluate each strain's EA utilization capacity. The W3110 strains (WT and Δ*eut*) behaved similarly in the control M9 medium with glycerol as a C source and NH$_4$Cl as a N source (Fig. 1B). However, only the WT strain grew in M9 glycerol EA B12, where EA was the only available N source (Fig. 1C). W3110 WT had comparable growth rates (Table 2) but significantly higher OD$_{600}$ after 24 h in M9 glycerol EA B12 than in M9 glycerol NH$_4$Cl (0.73 ± 0.01 vs 0.56 ± 0.04). BW25113 WT behaved like W3110 Δ*eut* and BW25113 *eut*+ behaved like W3110 WT (Fig. S1D and E), indicating that an uninterrupted *eut* operon is essential for EA utilization as a N source. Note that none of the strains grew in M9 EA B12 where EA was the only available N and C source (Fig. S1F). In M9 glycerol NH$_4$Cl with gentamicin (Gm), W3110 Δ*eut* pEut's growth rate was similar to that of the empty vector control (W3110 Δ*eut* pO) and the corresponding plasmid-bearing WT derivatives (Table 2; Fig.

**TABLE 2** Growth of *E. coli* K-12 W3110 strains grown in various M9 medium derivatives[a]

| | $\mu_{max}$ (h⁻¹) in: | | | |
|---|---|---|---|---|
| Strain | M9 glycerol NH$_4$Cl | M9 glycerol NH$_4$Cl IPTG | M9 glycerol EA | M9 glycerol EA IPTG |
| WT | 0.37 ± 0.01 | 0.37 ± 0.01 | 0.39 ± 0.01 | 0.39 ± 0.01 |
| Δ*eut* | 0.36 ± 0.03 | 0.35 ± 0.01 | 0.09 ± 0.06 | 0.03 ± 0.02 |
| WT pO | 0.35 ± 0.01 | 0.36 ± 0.01 | 0.38 ± 0.03 | 0.35 ± 0.01 |
| WT pEut | 0.33 ± 0.01 | 0.19 ± 0.01 | 0.37 ± 0.01 | 0.23 ± 0.01 |
| Δ*eut* pO | 0.35 ± 0.01 | 0.35 ± 0.01 | 0.12 ± 0.11 | 0.02 ± 0.01 |
| Δ*eut* pEut | 0.33 ± 0.01 | 0.19 ± 0.01 | 0.37 ± 0.01 | 0.25 ± 0.01 |

[a]These experiments were performed using a microplate reader. Data are shown as mean (*n* = 3 biological replicates) ± standard deviation.

1B). W3110 Δ*eut* pO did not grow in M9 glycerol EA B12 Gm (Fig. 1C) while W3110 Δ*eut* pEut utilized EA as sole N source with a maximal growth rate similar to that of W3110 WT pO (Table 2) but with a markedly lower final $OD_{600}$ (0.60 ± 0.01 vs 0.77 ± 0.01). The W3110 *eut* phenotype was thus at least partially recovered in the plasmid construct, even without IPTG induction, which indicates that the $P_{trc}$ promoter is leaky. When 20 µM IPTG was added at culture initiation, W3110 WT pEut and W3110 Δ*eut* pEut had significantly lower growth rates than the empty vector controls in both M9 glycerol $NH_4Cl$ Gm and M9 glycerol EA B12 Gm (Fig. 1D and E; Table 2). These results show that *eut* expression levels must be kept low to avoid any negative impact on growth and enable efficient EA utilization.

## Subcellular localization of the EutC enzyme

The EUT1 locus encodes a functional EA utilization pathway but does it truly drive the production of Eut BMCs *in cellula*? We first attempted to visualize *E. coli* Eut BMCs by fluorescence microscopy and EutC-GFP fusion. We focused on EutC because it natively carries a 20 amino-acid long N-terminal peptide (referred to as $EutC_{1-20}$ hereafter) that should ensure its encapsulation within BMCs (21, 22). To keep expression levels low, the synthetic gene was inserted at the SS9 safe chromosomal locus (30) (Table 1; Fig. 2A) in W3110 WT under the control of an anhydrotetracycline (aTc) inducible promoter [$P_{LtetO-1}$ (31)].

No GFP fluorescence emission was detected in the absence of aTc (Fig. S2A). In M9 glycerol $NH_4Cl$ with 8 ng·mL$^{-1}$ aTc, a faint cytosolic GFP signal was observed (Fig. 2B). In M9 glycerol EA B12 with 8 ng·mL$^{-1}$ aTc, multiple GFP puncta were observed moving rapidly around the cytosol (Fig. 2C; Movie S1; wider views in Fig. S3). These mobile GFP puncta are similar to those visualized by fluorescence microscopy for the *S. enterica* Eut BMC system (21, 32). There were 5.9 ± 1.1 GFP puncta per bacterium on average, a number that did not vary significantly upon increasing the aTc concentration up to 80 ng·mL$^{-1}$ (6.3 ± 1.2) (Fig. 2D and E). To investigate whether $EutC_{1-20}$ alone could drive the encapsulation of a heterologous protein, we also built a smaller cassette expressing the

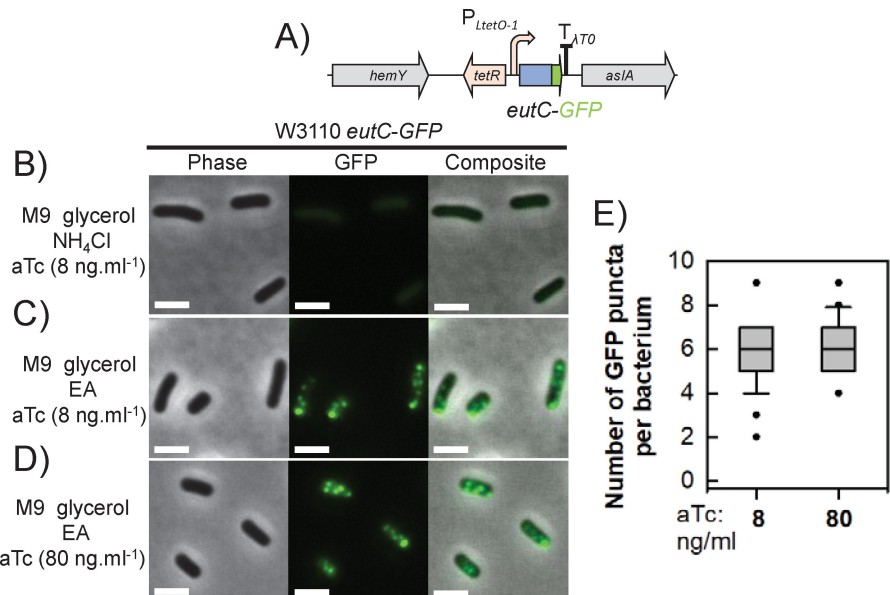

FIG 2 The chimeric protein EutC-GFP forms distinct puncta in an EA-containing medium. (A) Schematic representation of the $P_{LtetO-1}$::*eutC-GFP* expression cassette. This construct was inserted at the SS9 safe chromosomal locus in W3110 WT to produce W3110 *eutC-GFP*. (B to D) Representative micrographs showing phase contrast images, GFP fluorescence signals, and composite images from W3110 *eutC-GFP* grown in (B) M9 glycerol $NH_4Cl$ with 8 ng·mL$^{-1}$ aTc, (C) M9 glycerol EA B12 with 8 ng·mL$^{-1}$ aTc, and (D) M9 glycerol EA B12 with 80 ng·mL$^{-1}$ aTc. Scale bars indicate 2 µm. (E) Box plots indicating the number of GFP puncta per bacterium (*n* = 30 analyzed cells) for cultures grown in M9 glycerol EA B12 with 8 ng·mL$^{-1}$ or 80 ng·mL$^{-1}$ aTc.

EutC$_{1-20}$-GFP fusion protein (Fig. S2B). W3110 *eutC$_{1-20}$-GFP* behaved like W3110 *eutC-GFP* but with a stronger cytosolic GFP background (Fig. S2C, D and E), making it harder to distinguish the puncta. The diffuse GFP background in W3110 *eutC$_{1-20}$-GFP* likely came from non-encapsulated EutC$_{1-20}$-GFP fusions.

## Morphology of Eut BMCs *in cellula*

We first used a traditional TEM-based approach to evaluate the integrity of the Eut BMCs *in cellula*. No BMC-like structure was detected in W3110 WT growing in M9 glycerol NH$_4$Cl (Fig. S4A). In M9 glycerol EA B12, irregular polyhedral assemblies roughly 100 nm in diameter, with sharp edges and a well-defined delimiting layer were observed in some slices (Fig. S4B). Since these assemblies satisfy the morphological criteria for BMCs (33) we refer to them hereafter as Eut BMCs. These were present in about 1 in 50 bacterial slices. On LB medium supplemented with EA and vitamin B12 to induce *eut*, Eut BMCs were detected in about 1 in 10 bacterial slices (Fig. S4C) suggesting that more BMCs were present. Finally, W3110 Δ*eut* pEut was cultivated in LB containing IPTG to strongly induce the *eut* operon expression. Here, BMCs with various polyhedral geometries and dimensions were observed in most bacteria (Fig. S4D). Increasing the *eut* operon expression strength thus increased the number of BMCs per bacterium but had a negative effect on growth.

CryoET reconstructions of W3110 WT grown in M9 glycerol EA B12 revealed several cellular features including the inner and outer membranes, ribosomes, and Eut BMCs (Fig. 3A and B). Multiple Eut BMCs were observed per bacterium, surrounded by a thin layer (the BMC shell) with sharp edges and vertices (Fig. 3C and D). The shell often completely enclosed the core but structures resembling partial Eut BMCs were also present (arrow in Fig. 3D). Further experiments are required to determine whether these partial Eut BMCs correspond to (dis)assembling states or are cryoET (e.g., missing wedge) artifacts. The Eut BMC core was somewhat granular in appearance but more electron-dense than the surrounding cytoplasm, with several encapsulated components, likely proteins. This was particularly apparent in bacteria with clear cytosol after partial lysis (Fig. 3D). Overall, these *E. coli* Eut BMCs were more heterogeneous in size and shape than cyanobacterial carboxysomes (34). Tomograms of the W3110 Δ*eut* pEut strain cultivated in M9 glycerol EA B12 Gm IPTG as a control showed a higher number of Eut BMCs per bacterium compared with the WT (Fig. 3E and F; Fig. S5), but with similar shapes and dimensions. These first-ever cryoET images of Eut BMCs in a bacterium demonstrate that native *E. coli* Eut BMCs are properly formed.

## EA utilization requires minimal reorganization of the proteome

Since Eut BMCs are protein-based, we investigated how their production affects *E. coli* using label-free quantitative mass spectrometry-based proteomics. Samples were collected and compared between exponentially growing cultures of W3110 WT maintained aerobically in M9 glycerol EA B12 vs M9 glycerol NH$_4$Cl. Of the 1,999 quantifiable proteins, only 48 were significantly more abundant and 8 less abundant in M9 glycerol EA B12 compared with M9 glycerol NH$_4$Cl (Fig. 4A; Table S3). Two distinct modules were identified among these proteins by interaction network analysis using STRING v11.5 (35) (Fig. 4B).

Some of the *eut* encoded proteins in the first module were detected in M9 glycerol EA B12 only (EutHNPR), while the others were barely detectable in M9 glycerol NH$_4$Cl and strongly upregulated in M9 glycerol EA B12, with fold changes ranging from 5 (EutJ) to 325 (EutM) (Fig. 4A). This included enzymes linked to EA catabolism (EutAPQDEGBC) and the structural Eut BMC shell components (EutSMNLK). We calculated the abundance of each Eut protein in M9 glycerol EA B12 corrected for the number of theoretically observable tryptic peptides (iBAQ values [36]: Fig. 4C), to correct for differences in molecular weight and amino-acid sequences. Relative to the most abundant protein, EutQ, an acetate kinase (37), whose relative abundance was set to 100, the most abundant predicted shell components were EutM, EutL, EutS, EutK, and EutN with

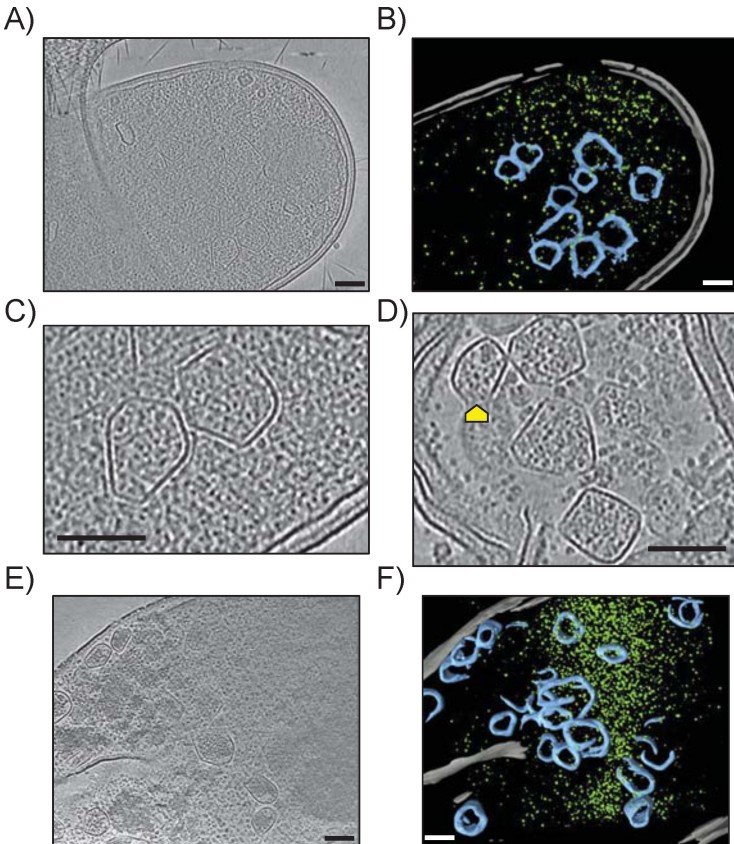

**FIG 3** *E. coli* K-12 W3110 produces well-formed Eut BMCs. (A) Representative electron cryotomogram slice of W3110 WT grown in M9 glycerol EA B12 and (B) associated 3D segmentation showing the cellular membranes in gray, the ribosomes in green, and the Eut BMCs in blue. (C and D) Expanded views of Eut BMCs illustrating their variety in size and shape. The yellow arrows in (D) point toward possible cryoET (missing wedge) artifacts or partially assembled BMCs. (E) Representative tomogram slice of W3110 Δeut pEut grown in M9 glycerol EA B12 Gm IPTG and (F) associated segmentation. Scale bars indicate 100 nm.

relative abundances of 96 ± 14, 39 ± 1, 11 ± 3, 8 ± 1, and 1 ± 1, respectively. The most abundant putative Eut BMC core enzymes besides EutQ were EutB, EutE, EutC, EutD, EutG, and EutP (a second acetate kinase), with relative abundances of 37 ± 17, 27 ± 10, 24 ± 9, 10 ± 3, 7 ± 1, and 7 ± 1, respectively.

Eleven ancillary proteins are encoded within the EUT1 locus (4, 16) (Fig. 1A; Table S2). We could quantify seven of them: the transketolase TktB, the transaldolase TalA, the malic enzyme MaeB, the N-acetylmuramoyl-L-alanine amidase AmiA, the putative acetyltransferase YpeA, the protein of unknown function YfeY as well as the porphyrinogen oxidase YfeX (Table S4). None of these proteins were differentially accumulated in M9 glycerol EA B12 compared to M9 glycerol NH$_4$Cl. Distinct regulatory mechanisms therefore control their production as compared to the *eut* operon encoded elements, i.e., the presence of EA and vitamin B12 does not induce the accumulation of the ancillary proteins. The four remaining ancillary EUT1 proteins were not detected, possibly because of low expression levels or a high hydrophobicity precluding their extraction/identification.

The second identified module consisted of proteins involved in maintaining intracellular N homeostasis (Fig. 4B), a majority of which were more highly expressed in M9 glycerol EA B12, including the ammonium transporter AmtB, its cognate regulator GlnK and a two-component system (GlnG-GlnL) controlling N assimilation (38, 39). The second module also contained enzymes catalyzing (de)amination reactions, such as asparagine synthetase (AsnAB), succinylglutamate semialdehyde dehydrogenase (AstCD), glutamine

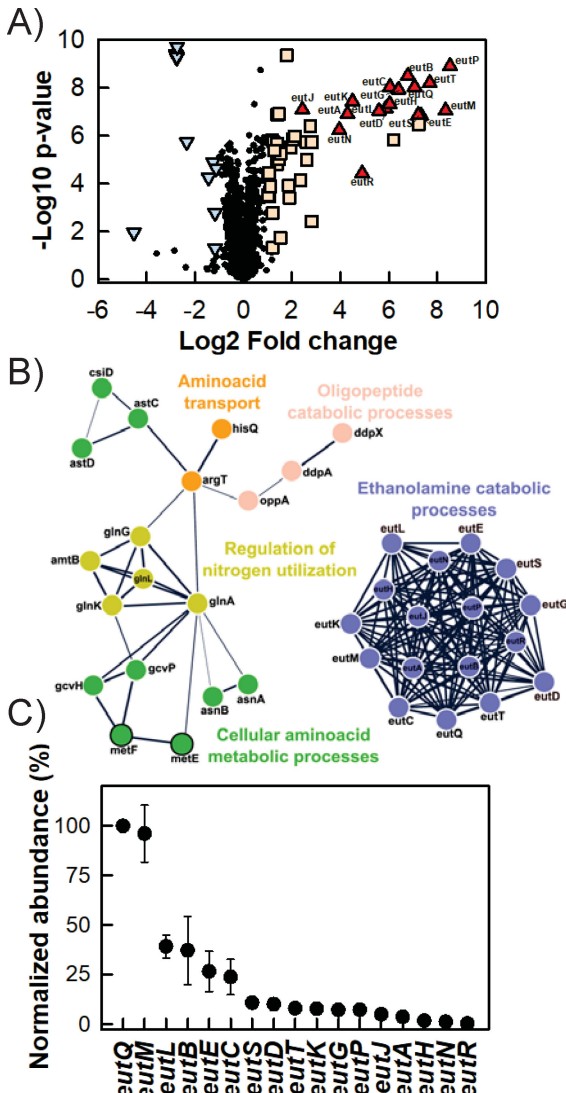

FIG 4 EA utilization requires minimal reorganization of the proteome. W3110 WT cultures were grown aerobically in M9 glycerol EA B12 vs M9 glycerol $NH_4Cl$. Samples were collected for label-free quantitative mass spectrometry-based proteomics analyses. (A) Volcano plot with Eut proteins (fold change > 2, $P < 0.05$) as red triangles, other upregulated proteins (fold change > 2, $P < 0.05$) as orange squares, downregulated proteins (fold change < −2, $P < 0.05$) as blue triangles, and unaffected proteins as black circles. (B) Protein-protein interaction network as determined using STRING v11.5. Only selected upregulated (white outline) and downregulated (black outline) proteins are shown. Thicker lines indicate greater confidence in the considered protein-protein interactions. (C) Normalized abundance (IBAQ score) of Eut proteins in M9 glycerol EA B12, using EutQ as the reference. Data shown are the mean of $n =$ 4 biological replicates, bars correspond to standard deviations.

synthetase (GlnA) and guanine deaminase (GuaD). Proteins involved in the uptake of extracellular dipeptides (DdpAX), polypeptides (OppA), and amino acids (ArgT, CycA, HisQ) were also present. Several regulators and pathways normally mobilized under N starvation conditions were therefore activated in M9 glycerol EA B12, probably due to the absence of extracellular N sources other than EA. Note that the abundance of cobalamin-independent homocysteine transmethylase (MetE) was seven times lower in M9 glycerol EA B12 than in M9 glycerol $NH_4Cl$. MetE normally participates in methionine biosynthesis but it can be replaced by MetH (cobalamin-dependent methionine synthase) in the presence of vitamin B12 (40).

Overall, these results indicate that the production of Eut BMCs did not have a pleiotropic effect, since only proteins involved in BMC synthesis and N metabolism were affected.

## Analysis of EA utilization by exometabolomics

To decipher the metabolic functioning of the Eut BMCs, we grew W3110 WT under aerobic conditions in M9 derivatives containing glycerol as a C source and either EA or $NH_4Cl$ as an N source (Fig. S6A and B). Vitamin B12 was again added to the EA-containing medium. The cultures reached similar maximal growth rates in both media (Table 3) but the biomass concentration after 12 h was higher with EA than with $NH_4Cl$ ($0.97 \pm 0.01$ $g_{DW} \cdot L^{-1}$ vs $0.72 \pm 0.01$ $g_{DW} \cdot L^{-1}$).

The exometabolome was analyzed by one-dimension proton nuclear magnetic resonance ($^1H$-NMR) and extracellular uptake and production fluxes were calculated from these data using PhysioFit (41) (Table 3; Fig. S6C and D). With $NH_4Cl$ as the sole N source, glycerol was fully assimilated and no by-products were detected. With EA as the sole N source, glycerol was consumed at a similar rate and EA was assimilated at a rate of $7.8 \pm 0.3$ $mmol \cdot g_{DW}^{-1} \cdot h^{-1}$. Ethanol ($q_{P)EtOH} = 2.6 \pm 0.1$ $mmol \cdot g_{DW}^{-1} \cdot h^{-1}$) and acetate ($q_{P)Ace} = 1.7 \pm 0.1$ $mmol \cdot g_{DW}^{-1} \cdot h^{-1}$) were excreted, and the acetate was re-consumed upon exhaustion of glycerol and EA, explaining the higher final biomass under this condition. Trace amounts of acetaldehyde were also detected, suggesting that some acetaldehyde leaked from the Eut BMCs and escaped from the cells. Acetate, ethanol, and acetaldehyde could all result from EA catabolism (20). Assuming NADH is recycled within the Eut BMCs, acetate and EtOH should be produced in similar amounts (19). Since the production rate of EtOH was significantly higher than that of acetate during exponential growth, this suggests either that some of the EA-derived acetate was used by cellular processes or that some of the EtOH was produced within the cytosol.

## Genome-scale predictions of EA utilization by *E. coli*

We next performed flux balance analysis [FBA (42)] on the same data to study the fate of EA in the Eut BMC and cytosol. We adopted the genome-scale model (GSM) of *E. coli* iML1515 (43) by compartmentalizing EA catabolism inside Eut BMCs, thereby ensuring that NADH and CoA-SH cofactors are recycled within the BMCs, while ATP is recycled in the cytosol (Fig. S7). The model was then constrained with the experimentally measured extracellular fluxes, assuming acetate and EtOH are produced in the BMCs. When maximizing biomass production, the maximal FBA-predicted growth rate was $0.72$ $h^{-1}$. This value is higher than observed experimentally, indicating that glycerol and EA utilization are suboptimal *in vivo*. We therefore constrained the growth rate to the experimental value (Table 3: $0.45$ $h^{-1}$) and defined ATP maintenance as the objective function (44) (Fig. S7). We also carried out flux variability analysis to identify the optimal solution and the range of fluxes that each reaction can carry while maintaining at least 99% of the objective.

The model predicted that Eut BMCs produce and release equimolar quantities of acetaldehyde, ethanol, and acetyl-P into the cytosol, each accounting for 33% of the C from EA (Fig. S7). Acetaldehyde can potentially escape from the cytosol into the extracellular medium, as reported previously (20), and/or be utilized in metabolism. All

TABLE 3 Growth parameters of *E. coli* K-12 W3110 WT grown aerobically in M9 medium containing glycerol as C source and $NH_4Cl$ or EA as N source

| Medium | Physiological parameter | | | | |
| --- | --- | --- | --- | --- | --- |
| | μ | $q_{S)glycerol}$ | $q_{S)ethanolamine}$ | $q_{P)acetate}$ | $q_{P)ethanol}$ |
| M9 gly $NH_4Cl$ | $0.44 \pm 0.02$ | $16.2 \pm 1.6$ | ND | ND | ND |
| M9 gly EA B12 | $0.45 \pm 0.01$ | $14.7 \pm 0.4$ | $7.8 \pm 0.3$ | $1.7 \pm 0.1$ | $2.6 \pm 0.1$ |

[a]The exponential growth rate μ is expressed in $h^{-1}$; the substrate uptake ($q_S$) and product formation ($q_P$) rates are expressed in $mmol \cdot g_{DW}^{-1} \cdot h^{-1} \cdot$DW, dry weight; ND, not detected. Data are shown as mean ($n = 3$ biological replicates) ± standard deviation.

the ethanol is excreted. Some of the acetyl-P is converted into acetate (21% of the C from EA) before excretion while the remaining portion can be converted into acetyl-CoA to fuel anabolism and thereby support growth (12% of the C from EA) (Fig. S7). While EA is thought to act only as a nitrogen source for *E. coli* K-12, genome-scale modeling thus predicts that it may also provide carbon. Moreover, the model indicates that the amount of EA-derived ammonium exceeds the N needs of *E. coli* for growth, and a metabolic steady state can thus only be achieved if some of the ammonium (38% of the N from EA) is excreted.

## $^{13}$C-metabolic flux analysis of EA and glycerol co-metabolism

To test the predictions of the GSM, we performed a $^{13}$C-metabolic flux analysis of W3110 WT grown in M9 medium containing $^{12}C_3$-glycerol, $^{13}C_2$-EA as well as vitamin B12 (Fig. 5) and quantified the time-course concentrations of labeled and unlabeled EtOH and acetate by $^{1}$H-NMR (Fig. 5A and B). EtOH was virtually fully labeled (4.66 ± 0.44 mM $^{13}C_2$-EtOH vs 0.47 ± 0.03 mM $^{12}C_2$-EtOH after 10 h, Fig. 5B), demonstrating that about 90% was produced from EA. The residual $^{12}C_2$-EtOH may have come either from incomplete encapsulation of Eut enzymes (e.g., EutE converting $^{12}C_2$-acetyl-CoA into $^{12}C_2$-acetaldehyde within cytosol and EutG yielding $^{12}C_2$-EtOH), from cytosolic acetyl-CoA (produced from unlabeled glycerol) entering misassembled BMCs, or from weak cytosolic conversion of glycerol to ethanol through alternative cytosolic pathways. In contrast, only half of the acetate pool was labeled, indicating that EA and glycerol contributed equally to acetyl-P synthesis and thus to acetate production. The extracellular ammonium concentration, measured by $^{1}$H-NMR, increased from 0.4 ± 0.1 mM after 3 h to 4.4 ± 0.1 mM after 11 h (Fig. 5B).

Finally, we built a dynamic isotopic model to quantify *in vivo* fluxes within and around the Eut BMCs by fitting the dynamics of all (labeled and unlabeled) exometabolites (Fig. 5C; Table S5). This model includes a coarse-grained representation of the glycolytic conversion of glycerol to acetyl-P and of the conversion of acetyl-P into acetate or its utilization elsewhere in metabolism for biomass synthesis, as suggested by the GSM. This model fits the data satisfactorily (Fig. S8), supporting the validity of the assumed network topology. Here, the Eut BMCs released more ethanol (Table S5; $v_{EutG}$ = 2.7 ± 0.1 mmol·$g_{DW}^{-1}$·$h^{-1}$, 37% of the C from EA) and acetyl-P ($v_{EutD}$ = 2.7 ± 0.1 mmol·$g_{DW}^{-1}$·$h^{-1}$, 37% of the C from EA) than acetaldehyde ($v_{BMC)acetaldehyde}$ = 1.7 ± 0.1 mmol·$g_{DW}^{-1}$·$h^{-1}$, 26% of the C from EA). Consistent with the labeling data, glycerol, and EA contributed equally to the cytosolic acetyl-P pool (with $v_{glycolysis}$ = 2.5 ± 0.1 mmol·$g_{DW}^{-1}$·$h^{-1}$). About 32% of the acetyl-P pool was excreted as acetate ($q_{P)Acetate}$ = 1.6 ± 0.1 mmol·$g_{DW}^{-1}$·$h^{-1}$; 12% of the C from EA) and the rest fuelled growth ($v_{Pta}$ = 3.5 ± 0.1 mmol·$g_{DW}^{-1}$·$h^{-1}$; 25% of the C from EA). To confirm the significant anabolic utilization of carbon derived from EA, we measured the carbon isotopologue distributions of proteinogenic amino acids (Fig. 5D). In keeping with the high $^{13}$C-enrichment of the cytosolic acetyl-P pool (and therefore also the acetyl-CoA pool) predicted by the isotopic model (Fig. S9), amino acids derived from the TCA cycle (i.e., Arg, Glu, Asp, Thr, Lys, Ile) had high fractions of heavy isotopologues. In contrast, amino acids produced from intermediates of the glycolytic and pentose phosphate pathways (i.e., Gly, Ser, His, Tyr, Phe, Ala, Val) had much lower $^{13}$C enrichment, pointing to an absence of neoglucogenic flux under the investigated conditions. Carbon derived from EA thus mainly entered central metabolism through the TCA cycle.

Regarding N metabolism, the isotopic model confirmed the ammonium overflow predicted by the GSM, with an ammonium production rate of 0.9 ± 0.2 mmol·$g_{DW}^{-1}$·$h^{-1}$ (Fig. 5C; Table S5; 12% of the N from EA). This value is slightly lower than the optimal FBA-predicted value (3.0 mmol·$g_{DW}^{-1}$·$h^{-1}$; Fig. S7), possibly because of an underestimation of the quantity of N required to form biomass (which was determined during growth on glucose and ammonia as sole C and N sources, respectively [43]).

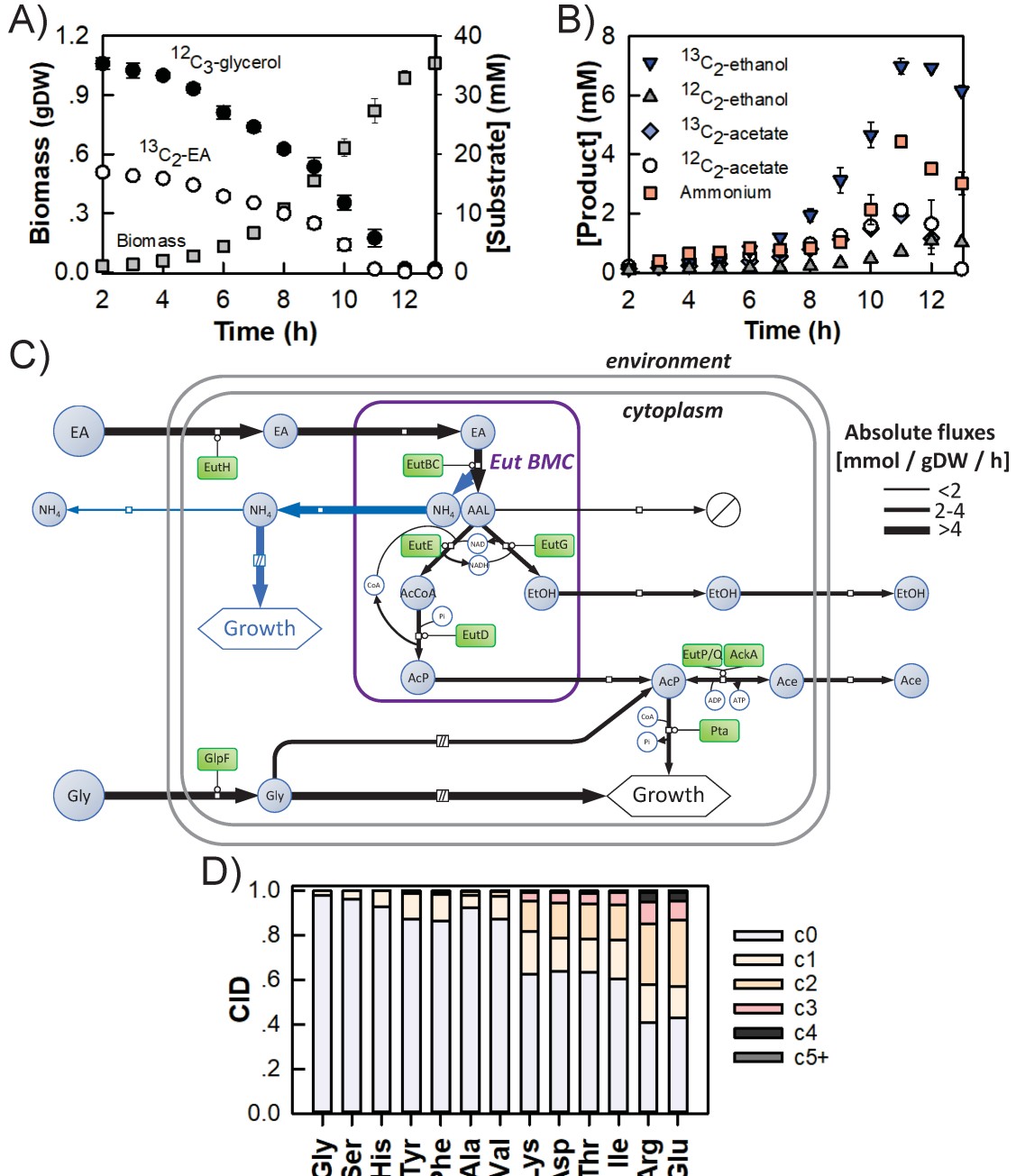

**FIG 5** The *E. coli* K-12 W3110 Eut BMCs turn EA into usable C and N sources. W3110 WT was grown aerobically in an M9 medium containing $^{12}C_3$-glycerol and $^{13}C_2$-EA. (A) Biomass accumulation (grams dry weight [$g_{DW}$]) and substrate consumption (mM); (B) product excretion. The substrates and products were quantified by $^1$H-NMR. Data shown are the average of $n = 3$ biological replicates, bars correspond to standard deviations. (C) Experimental and predicted fluxes through EA and glycerol metabolisms using an isotopic model. Absolute flux values are summarized in Table S4. Blue-filled circles indicate metabolites and green-filled rectangles indicate proteins. Unidentified sinks are indicated as barred white circles. (D) Carbon isotopologue distribution (CID) within proteinogenic amino acids from biomass samples harvested during exponential growth, showing the intracellular incorporation of $^{13}$C derived from EA. AAL, acetaldehyde; AcCoA, acetyl-CoA; Ace, acetate; AcP, acetyl-phosphate; EtOH, ethanol; $NH_4$, ammonium.

These isotope labeling experiments thus confirm the main predictions of the GSM, namely that EA can be used as a C source for growth and that N overflow occurs under these conditions.

## DISCUSSION

The presence of functional Eut BMCs in *E. coli* has been suggested but never clearly demonstrated. Focusing on the non-pathogenic laboratory strain *E. coli* K-12, our interdisciplinary results provide deep molecular and functional insights into this native Eut BMC system and its importance for EA utilization.

We were able to visualize *E. coli* Eut BMCs *in cellula* by TEM and even more clearly by cryoET. The Eut BMCs produced by WT *E. coli* K-12 W3110 were mostly well formed with an electron-dense core separated from the surrounding cytoplasm by a continuous shell layer. Dark particles, presumably Eut enzymes, were present in the core. No array-like particle arrangements were observed, contrary to what has been reported for carboxysomes (45–49), but similar to observations of the Pdu BMCs of *Acetonema longum* (50). While recent efforts have allowed the internal organization of carboxysomes to be mapped out, much less is known of metabolosomes. Working on isolated Eut BMCs would facilitate downstream structural characterization; however, our attempts to purify *E. coli* Eut BMCs using protocols established for Pdu BMCs (51) or α-carboxysomes (52) failed (data not shown). The protein-protein interactions that allow Eut BMCs to assemble and/or persist *in vivo* do not withstand extracellular conditions (53). Similar difficulties have previously been reported in *S. enterica* (51) and to our knowledge intact Eut BMCs have never been purified from any bacterial species so far, suggesting that Eut BMCs are intrinsically less stable than Pdu BMCs or carboxysomes.

We used fluorescence microscopy to observe Eut BMCs *in vivo*. EutC$_{(1-20)}$-GFP formed discrete puncta in M9 glycerol EA B12 only so upon *eut* induction, i.e., when all the other Eut components were present. EutC$_{(1-20)}$-GFP was likely encapsulated inside the Eut BMC core under these conditions. Unlike fluorescently tagged carboxysomes, which align along the cell axis (48), the EutC$_{(1-20)}$-GFP puncta moved freely around the cytoplasm. This sort of mobility has also been observed for SeEutC$_{1-19}$-EGFP encapsulated in *S. enterica* Eut BMCs *in vivo* (21, 32). Eut BMCs may interact with the bacterial cytoskeleton through shell-coating proteins, somewhat like Pdu BMCs do with PduV (54). EutP and PduV share some sequence homology (e.g., 28% primary structure identity and 54% similarity between EutP from *E. coli* K-12 and PduV from *S. enterica* LT2), both containing a RAS-like GTPase superfamily domain (Pfam10662). EutP may hence play a role in positioning the Eut BMCs, but this hypothesis remains to be validated experimentally. How Eut BMC subcellular dynamics affect their metabolic functioning or inheritance by daughter cells remains unclear. An EutK-mCherry fusion has recently been used to visualize *Listeria monocytogenes* Eut BMCs *in vivo* (55), which appeared as immobile red puncta. Future work should include the colocalization of core (e.g., EutC) and shell (e.g., EutK) elements to further validate the location of the fluorescent reporters.

How many Eut BMCs do the *E. coli* K-12 W3110 cells contain under our culturing conditions and how is the number of Eut BMCs regulated? Assuming that each GFP punctum corresponded to a single BMC, our data indicate that there were 6 Eut BMCs per bacterium, but this count needs to be confirmed by higher resolution approaches such as correlated light and electron microscopy (56). In comparison, around 3–4 GFP puncta per bacterium were observed in *S. enterica* after labeling Pdu BMCs with GFP (57). TEM observations indicate that chemo-autotrophic bacteria contain 3 to 80 carboxysomes per cell (34, 48). The number of Eut BMCs likely depends on culture conditions (9): resource availability and specific regulatory networks (e.g., control by catabolic repression (10) may modulate expression of the *eut* operon. Even with 5 to 100 BMCs per *E. coli* W3110 WT, Eut BMCs would occupy just 0.2%–4% of the total cytosolic volume.

This study also shows that the ability of *E. coli* K-12 to utilize EA as an N source hinges on the *eut* operon, with minor reorganizations of the rest of the proteome. However, expressing *eut* from even a low copy plasmid impaired growth. The underlying physiological cause is most likely the associated metabolic burden because slower growth (i) also occurred in a medium containing NH$_4$Cl as the N source (indicating that the impairment is not simply due to impaired EA catabolism) and (ii) was more pronounced after inducing P$_{trc}$ with IPTG. Overexpressing the *eut* operon increased the

number of Eut BMCs per bacterium observed by electron microscopy, up to several hundred under certain conditions (e.g., LB IPTG). The same effect (i.e., more Eut BMCs and slower growth) has previously been observed in *Enterococcus faecalis* upon suppressing a negative *eut* regulator (58). As observed for other proteins (59, 60), fine-tuned regulation of the *eut* operon is necessary to achieve the desired metabolic function at a manageable metabolic cost.

The protein complement of Eut BMCs remains unclear; however, our relative quantification of *E. coli* Eut proteins *in vivo* can be used to make an educated guess. The most abundant candidate shell component was hexameric (EutM), the second-most was trimeric (EutL) and the most dilute was pentameric (EutN), reminiscent of the Pdu BMC system in *S. enterica* (61). This is consistent with EutM and EutL forming the facets and edges of the polyhedral shell together with other minor building blocks (i.e., EutS and EutK) (17, 62). As a pentameric vertex protein (17, 63), EutN is less abundant than the other shell components. Among candidate core enzymes, the large (EutB) and small (EutC) subunits of EAL were roughly equimolar, consistent with the holoenzyme's predicted oligomeric state [$(EutB_2EutC_2)_3$] (64). NADH generating EutE acetaldehyde dehydrogenase was another major putative core component as were Eut enzymes involved in cofactor recycling (EutG for $NAD^+$ and EutD for CoA-SH).

EutQ was found to be highly abundant, ten times more abundant than EutP, the other *eut*-encoded acetate kinase (37). We think that EutQ and EutP are localized outside the Eut BMC (i.e., on the shell surface or soluble in the cytosol) so that any ATP generated upon EA-derived acetyl-P consumption can feed into cellular metabolism. Moreover, EutP must be localized on the shell surface to interact with cytoskeleton elements. However, our results do not allow any conclusions to be drawn about the subcellular localizations of EutQ and EutP: either or both enzyme(s) could also be encapsulated within BMCs *in vivo*. EutQ has been reported to interact with EutM *in vitro* and to facilitate the formation of multiple Eut BMCs per bacterium (32). Since ATP probably cannot transit through the pores in the shell (19), any ATP produced in the core by EutQ and EutP when converting acetyl-P into acetate would have to be recycled internally, e.g., to reactivate EAL after it undergoes catalytic inactivation (65, 66). During the finalization of the present manuscript, a study was released showing the cytosolic localization of EutP and the encapsulation of EutQ within Eut BMCs in *S. enterica* LT2 (67). EutQ moreover seems to play a central role in connecting the Eut BMC core and shell elements (67).

To our knowledge, the flux map reported here is the first to connect the Eut BMC with central metabolism *in cellula*. Our results indicate that *E. coli* Eut BMCs leak non-negligible amounts of acetaldehyde (up to 33% of the C from EA). In comparison, about 13% and 28% of the C from 1,2-PD was found to be lost as propionaldehyde by the Pdu BMCs of *S. enterica* (68) and *Propionobacter freudenreichii* (69), respectively. Leakage may thus be a common feature of metabolosomes. However, our cultures were performed under aerobic conditions with shaking, which favors gas exchanges and acetaldehyde evaporation (20). These conditions are different from those encountered within the gut environment, where only limited quantities of oxygen diffuse through the mucosal surface to be consumed by facultative anaerobes such as *E. coli* (70). Hence, less acetaldehyde losses may occur *in vivo* within the mammalian host. We did not observe any deleterious effect of acetaldehyde leakage on growth, suggesting that the concentration of acetaldehyde remained below the toxicity threshold in the cytosol (20). In our model, the remaining acetaldehyde is converted inside the BMC core into equimolar quantities of ethanol and acetyl-P before diffusing out through shell pores. Some of the EA-derived C then enters central metabolism, mainly through the TCA cycle, to fuel growth. This result is in line with a recent report demonstrating that *E. coli* assimilates acetate (here derived from ethanolamine) with glycolytic substrates when the glycolytic flux is low, such as on glycerol (71), to feed anabolism. But why then is *E. coli* K-12 W3110 WT unable to grow with EA as the sole C source? Perhaps the flux of EA assimilation simply did not meet cellular housekeeping demands. Additional comparisons with other

*E. coli* strains, including some capable of utilizing EA as the sole C source (10, 12), are required to identify the underlying mechanisms.

Could some ancillary EUT1 proteins play a role in EA utilization? We detected 7 out of the 11 ancillary EUT1 proteins by proteomics (TktB, TalA, MaeB, AmiA, YpeA, YfeY, and YfeX), none of them being differentially accumulated in M9 glycerol EA B12 vs M9 glycerol $NH_4Cl$. YpeA and YfeY have no known biological functions. AmiA participates in septum formation during cell division (72). YfeX is a peroxidase that acts on por-phyrinogens to convert them into porphyrins (73); *yfeX* is upregulated under anaerobic conditions and may participate in respiratory complexes recycling. The contributions of AmiA and YfeX to EA catabolism, if any, appear unclear. Under certain conditions, MaeB, TktB, and TalA, on the other hand, may funnel some EA-derived C through gluconeogenesis and the pentose phosphate pathway. Malic enzyme activity plays an important metabolic role by connecting the TCA cycle with gluconeogenesis (74). The *maeB* expression is moreover induced when *E. coli* grows on acetate (75). MaeB may thus be key when *E. coli* grows using the EA-derived acetyl-P (or acetate) as its sole C source: after conversion into acetyl-CoA, these EA-derived compounds first enter central metabolism through the TCA cycle/glyoxylate shunt. MaeB would then direct some EA-derived C towards gluconeogenesis. TktB and TalA both participate in the pentose phosphate pathway (PPP) (76). TktB and TalA would next direct some of the EA-derived C through the PPP, allowing the generation of multiple metabolites that are essential for biomass formation (erythrose-4P, ribulose-5P…). TalA, like other aldolases (77), may also be able to condensate some EA-derived acetaldehyde (leaking from the Eut BMCs) with glyceraldehyde-3P to produce 1-deoxy-D-xylulose 5-phosphate, a precursor for the synthesis of vitamin B6. The ancillary EUT1 proteins may hence funnel some EA-derived C for the accumulation of biomass. However, in the present study, we had to perform our experiments in M9 glycerol EA B12 since *E. coli* K-12 W3110 does not grow on M9 EA B12 (i.e., with EA as the sole C/N source). In our medium, glycerol provides the most C for both glycolysis and the pentose phosphate, as confirmed by the labeling patterns of proteinogenic amino acids. The ancillary EUT1 proteins implication should therefore be re-assessed in *E. coli* strains that grow using EA as their C and N source, under more physiologically relevant conditions.

By integrating our microscopy and fluxomics data, we calculated the mean EA conversion flux through an individual *E. coli* K-12 Eut BMC *in cellula* (calculations in Data S2): the value is close to 0.300 fmol·BMC$^{-1}$·h$^{-1}$ (or 0.600 fmol C·BMC$^{-1}$·h$^{-1}$). This is very close to the mean flux of 1,2-PD conversion through an individual *S. enterica* Pdu BMC *in cellula* (0.350 fmol·BMC$^{-1}$·h$^{-1}$ or 1.050 fmol C·BMC$^{-1}$·h$^{-1}$) calculated from previous experimental and modeling data (68, 78). The carboxylation rate through an individual carboxysome varies from 0.03 to 4.82 fmol C·BMC$^{-1}$·h$^{-1}$ depending on conditions (79). These BMC types seem to operate at similar overall turnover rates, despite having different enzymatic core contents. Although they occupy an almost negligible fraction of the cytosolic volume, our model predicts that Eut BMCs act as efficient nanobioreactors with total C fluxes comparable to those of other major cellular metabolic processes (e.g., a glycolytic flux of 0.72 fmol·cell$^{-1}$·h$^{-1}$ or 2.16 fmol C·cell$^{-1}$·h$^{-1}$ here).

Finally, we also found that EA served as an N source, with strong modifications in the levels of proteins involved in nitrogen metabolism. The ammonium generated by EA catabolism was mostly used for biomass production but the rest was excreted. The N:C molar ratio of EA (1:2) is higher than in the elemental composition of *E. coli* (1:4) (80), which explains why our cultures acted as net ammonium producers. Moreover, given that glycerol was the primary C source in M9 glycerol EA B12, the proportion of excreted ammonium would likely be higher in other media with less organic C (e.g., M9 EA). Another study, released during the finalization of the present manuscript, showed that *E. coli* Nissle 1917 also displays an ammonium overflow when grown on M9 EA B12 while *S. enterica* does not (81). This may have important physiological implications in the gut microbiome environment. Do EA-utilizing bacteria release ammonium in the gut,

making it available for surrounding micro-organisms? EA utilization may influence the composition of complex bacterial communities through C and N overflows.

## Conclusion

The native *E. coli* Eut BMC has so far been overlooked by recent efforts to explore the natural diversity of BMC structures and functions. Our interdisciplinary approach shows that the laboratory strain *E. coli* K-12 W3110 produces well-formed Eut BMCs. These Eut BMCs act as orthogonal modules in the cytosol, turning EA into usable N and C substrates for biomass formation, but with significant metabolic overflow. However, some questions remain unanswered as to how the Eut BMCs operate, particularly regarding the subcellular localization of certain Eut proteins (e.g., EutQ) and cofactor recycling (e.g., ATP). Several ancillary EUT1 proteins may be important for EA utilization but the hypothesis still awaits direct experimental validation. The existence of a native subcellular compartmentalization system shows that *E. coli* has a more complex metabolic organization than originally perceived.

## MATERIALS AND METHODS

### Bacterial strains and culture conditions

#### Media preparation

Cloning experiments were performed in lysogeny broth (LB). For the physiological characterizations, a nitrogen-free M9-medium was first assembled containing (final concentration in the medium): 17.4 $g \cdot L^{-1}$ $Na_2HPO_4 \cdot 12H_2O$, 3.03 $g \cdot L^{-1}$ $KH_2PO_4$, 0.51 $g \cdot L^{-1}$ NaCl, 0.49 $g \cdot L^{-1}$ $MgSO_4$, and 4.38 $mg \cdot L^{-1}$ $CaCl_2$. Thiamine hydrochloride (100 $mg \cdot L^{-1}$) was also included, as well as 0.1% (vol/vol) of a trace element solution (final concentration in the medium: 15 $mg \cdot L^{-1}$ $Na_2EDTA \cdot 2H_2O$, 4.5 $mg \cdot L^{-1}$ $ZnSO_4 \cdot 7H_2O$, 0.3 $mg \cdot L^{-1}$ $CoCl_2 \cdot 6H_2O$, 1 $mg \cdot L^{-1}$ $MnCl_2 \cdot 4H_2O$, 1 $mg \cdot L^{-1}$ $H_3BO_3$, 0.4 $mg \cdot L^{-1}$ $Na_2MoO \cdot 2H_2O$, 3 $mg \cdot L^{-1}$ FeSO4 $\cdot$ 7H$_2$O, and 0.3 $mg \cdot L^{-1}$ $CuSO_4 \cdot 5H_2O$). In M9 glycerol ammonium, $NH_4Cl$ (1.07 $g \cdot L^{-1}$ or 20 mM) was added as the nitrogen source, and glycerol (2.76 $g \cdot L^{-1}$ or 30 mM) as the organic carbon source. In M9 glycerol EA B12, glycerol was the main organic carbon source (2.76 $g \cdot L^{-1}$ or 30 mM), ethanolamine was added as the nitrogen source (1.22 $g \cdot L^{-1}$ or 20 mM; from a stock solution at 122 $g \cdot L^{-1}$ adjusted to pH 7 with HCl 12N) and vitamin B12 (B12) was systematically included (200 nM). All media components were autoclaved except for thiamine hydrochloride, the trace element solution, $NH_4Cl$, and ethanolamine which were filter-sterilized instead (Minisart 0.2 mm syringe filter, Sartorius, Germany). Antibiotics were added to both solid and liquid media according to each strain's resistance profile (ampicillin [Amp] at 100 $\mu g \cdot mL^{-1}$, kanamycin [Km] at 50 $\mu g \cdot mL^{-1}$, spectinomycin [Sp] at 50 $\mu g \cdot mL^{-1}$, and/or gentamicin [Gm] at 10 $\mu g \cdot mL^{-1}$). IPTG (final concentration 20 $\mu M$) or anhydrotetracycline (aTc) (final concentration 8 $ng \cdot mL^{-1}$ or 80 $ng \cdot mL^{-1}$) was also incorporated when indicated. All chemical products were purchased from Sigma-Aldrich (France) unless otherwise specified.

#### Strain cultivation

*E. coli* K-12 W3110 WT and several mutant derivatives were used (Table 1). All the strains were cryopreserved at −80°C in LB with 25% (wt/vol) glycerol. The strains were streaked onto LB agar plates and incubated at 37°C for 16 h. An isolated single colony then served to inoculate 2 mL of LB medium before culturing for 8 h at 37°C under 200 rpm orbital shaking (Inova 4230, Brunswick Scientific, United States). The optical density at 600 nm ($OD_{600}$) was measured with a Genesys 6 spectrophotometer (Thermo Fisher Scientific, United States). The preculture was next diluted into a baffled shake flask (250 mL) containing 50 mL of modified M9 medium, aiming for a starting $OD_{600}$ of 0.07.

After 16 h of incubation at 37°C under 200 rpm shaking, the bacteria were collected by centrifugation (4,000 $\times$ $g$ at room temperature for 3 min) and washed once using the

modified M9 medium. Finally, the bacteria were inoculated into a baffled flask (250 mL) containing 50 mL of modified M9 medium, again at a starting $OD_{600}$ of 0.07. Cultures were performed at 37°C under 200 rpm shaking to proceed with the physiological characterization. The following equation was employed to convert $OD_{600}$ values into corresponding biomass dry weight: $g_{DW} = 0.37 \times OD_{600}$.

When mentioned, cultures were made in a plate format instead. In this case, transparent flat-bottom 96 well plates (Sarstedt, Germany) covered with lids and containing 100 µL medium per well were used. The culturing scheme was similar to that described for shake flasks. A CLARIOStar Plus (BMG LabTech, Germany) plate reader apparatus allowed for maintaining the cultures at 37°C under 200 rpm double orbital shaking. Absorbance at 600 nm was measured every 10 min to estimate growth. Results were analyzed with the MARS Data Analysis software (BMG LabTech, Germany).

## Molecular biology

### Cloning procedures

Polymerase chain reactions (PCRs) for cloning purposes were performed using the high-fidelity Phusion DNA Polymerase (NEB, France). PCR products were purified with the NucleoSpin PCR Clean Up kit (Macherey Nagel, Germany), quantified with a NanoDrop 2000 Spectrophotometer (Thermo Fisher Scientific, France), and assembled by In-Fusion (TaKaRa, Japan). After purification with the NucleoSpin Plasmid kit (Macherey Nagel, Germany), the generated plasmids were verified by Sanger Sequencing.

### CRISPR-Cas9 genome editing

The system described by Jiang et al. was utilized in this study (27). To generate the Δ*eut* strain, a pTargetF plasmid for *eutG* (pTargetF_eutG) was first constructed. The linear backbone from pTargetF was amplified with primers F_bb1/R_bb1 (see Table S1 for primer sequences). The *eutG* targeting protospacer (PAM shown in bold: CGGCACACCTT CGGTCAATG**CGG**) was amplified with primers F_eutGgRNA/R_eutGgRNA. The generated amplicons were then assembled by In-Fusion to generate pTarget_eutG (Sp$^R$). Next, the 500 nt situated directly upstream of the P*eut* promoter (upper homology arm) were amplified from purified *E. coli* K-12 W3110 WT gDNA with F_UA1/R_UA1. The same was achieved for the 500 nt downstream of *eutR* (lower homology arm) with F_LA1/R_LA1. The backbone from pUC19 was amplified with F_bb2/R_bb2 and all three fragments were assembled by In-Fusion to yield pHD_Δeut (Amp$^R$). The linear fragment containing both homology arms was amplified from pHD_Δeut using F_HD1/R_HD1. pTarget_eutG as well as the produced linear fragment were finally electroporated into pCas-contain-ing *E. coli* K-12 W3110 WT bacteria to obtain chromosomally edited Δ*eut* strains. After genotyping, the edited mutants were cured of pTarget_eutG and pCas as described previously.

The *SS9* targeting protospacer (TCTGGCGCAGTTGATATGTA**AGG**) was cloned by In-Fusion into pTargetF using primers F_SS9gRNA/R_SS9gRNA and F_bb1/R_bb1, yielding pTargetF_SS9. The 1,000 nt upstream of *SS9* were amplified from purified *E. coli* K-12 W3110 WT gDNA with F_UA2/R_UA2, including mutations to suppress the above-mentioned PAM (i.e., AGG to CAA), the 1,000 nt downstream of *SS9* with F_LA2/R_LA2. The EutC-GFP and EutC$_{1-20}$-GFP encoding cassettes (P$_{LtetO-1}$ promoter, T$_{LT0}$ terminator) were synthesized and cloned with flanking the SS9 homology arms into pUC19, yielding pHD_SS9_eutC-GFP and pHD_SS9_eutC$_{1-20}$-GFP.

### Assembly of pEut_WT

The linear backbone from pO (a modified version of pSEVA661 bearing a *lacI$^q$* expression cassette and the P$_{trc}$ promoter cloned in its multiple cloning site) was amplified with primers F_bb3/R_bb3 (see Table S1). The complete *eut* operon was amplified from isolated *E. coli* K-12 W3110 WT gDNA using F_eut1/R_eut1, F_eut2/R_eut2 as well as F_eut3/R_eut3. The generated amplicons were assembled by In-Fusion to produce

the low-copy plasmid pEut_WT (Gm$^R$). Next-generation DNA sequencing on an Ion S5 System (Thermo Fisher Scientific, France) was performed at the GeT_Biopuces platform (TBI, INSA Toulouse, France) to verify that the plasmid was properly assembled.

## Microscopy observations

### Epifluorescence microscopy

Samples (equivalent to 0.6 µL culture at $OD_{600}$ = 1) were collected after reaching the stationary phase and deposited onto GeneFrames (ThermoFisher) containing solidified C-/N-free M9 medium with 1% agarose. Phase contrast and fluorescence microscopy were performed at room temperature using an automated inverted epifluorescence microscope (Nikon Ti-E/B) equipped with the "perfect focus system" (PFS, Nikon), a phase contrast objective (Plan Apo 100× Oil Ph3 DM NA1.4), a Lumencor SpectraX Light Engine as the illumination source (Ex: 475/34 for GFP), Semrock Brightline multiband dichroic filters (FF409/493/573/652-Di02 for GFP), Semrock emission filters (Em: 536BP40 for GFP), and a Flash4.0 sCMOS camera (Hamamatsu). Fluorescence images were captured and processed using Nis-Elements AR software (Nikon) as well as Fiji (82). To visualize the mobile fluorescent puncta, pictures were taken every 5 s for 30 s. The puncta were counted manually within $n$ = 30 bacterial cells per sample.

### TEM visualization

Samples (equivalent to 1 mL of culture at $OD_{600}$ = 1) were collected upon reaching the stationary phase by centrifugation at 3,000 × $g$ for 3 min. Pellets were resuspended in fixation solution (2.5% glutaraldehyde, 0.1 M cacodylate pH 7.4 as well as 0.04% (wt/vol) ruthenium red) and incubated for 16 h at 4°C. Samples were post-fixated in 1% osmium tetroxide, dehydrated stepwise in ethanol, and eventually embedded in the Embed 812 resin (Electron Microscopy Sciences) with a Leica AMW automated device. Thin sections (70 nm width) were stained (solution containing 3% uranyl acetate in 50% ethanol as well as Reynold's lead citrate) and observed using an HT 7700 Hitachi transmission electron microscope (accelerating voltage 80 kV, CCD AMT XR41 camera).

### cryoET visualization

Samples (equivalent to 1 mL of culture at $OD_{600}$ = 1) were collected upon reaching the stationary phase and centrifuged at 3,000 × $g$ for 3 min. Supernatants were discarded. Pellets were washed twice in 1× PBS buffer (pH 7.4) and finally resuspended in 50 µL of 1× PBS (pH 7.4).

For grid preparation, 3.4 µL of sample was deposited onto glow-discharged lacey carbon grids and placed in the thermostatic chamber of a Leica EM-GP automatic plunge freezer, set at 20°C and 95% humidity. The excess solution was removed by blotting with Whatman no. 1 filter paper for 2.5 s, and the grids were immediately flash-frozen in liquid ethane at −185°C.

For cryo-electron tomography, tilt-series were acquired on a Talos Arctica (Thermo Fisher Scientific) operated at 200 kV in parallel beam condition using either a K2 (for mutant cells) or K3 (for WT condition) Summit direct electron detector and a BioQuantum energy filter (Gatan Inc.) operated in zero-loss mode with a slit width of 20 eV. For WT cells, data collection was carried out using Tomo software (Thermo Fisher Scientific), at a nominal magnification of ×31,000 with a calibrated pixel size of 2.78 Å. For mutant cells, a nominal magnification of ×49,000 was used with a calibrated pixel size of 2.84 Å. In both conditions, tilt series were acquired following the dose symmetric scheme (83) between +50° and −50° with a 2° tilt increment and a defocus range between −10 and −12 µm. Each tilt image was acquired in electron counting mode with a cumulative electron dose of 140e$^-$/A$^2$ fractionated into 10 frames.

For tomogram reconstruction and segmentation, movie frames were aligned using MotionCor2 (84) and tilt series were mutually aligned by using 10 nm gold particles as fiducial markers and the 3D volumes (tomograms) were reconstructed with weighted

back projection using IMOD v.4.11.16 (85) software packages with a binning factor of 4. Tomograms were finally denoised and contrast-enhanced, boosting the signal-to-noise ratio, using Topaz (86) and non-linear anisotropic diffusion filtering, respectively. Individual objects in tomograms are then segmented using EMAN2 TomoSeg (87), and the surfaces of each segmented object are generated and visualized in ChimeraX (88).

## Proteomics analysis

*E. coli* K-12 W3110 WT cultures were grown at 37°C under 200 rpm orbital shaking, in 250 mL flasks containing 50 mL M9 glycerol EA B12. Upon reaching $OD_{600} = 1$, samples (40 mL) were collected and centrifuged for 5 min at 4,000 × *g*. The supernatant was discarded, and the pellet was resuspended in 3 mL lysis buffer (100 mM triethylammonium bicarbonate [TEAB] buffer [pH 8.5] with 2.5% [wt/vol] SDS) at 4°C. Bacteria were lyzed by sonication (Fisherbrand sonicator equipped with a microprobe: 20% power, 20 s on, 30 s off, the cycle repeated twice). After centrifugation for 8 min at 13,000 × *g*, the supernatant was collected. The protein concentration was determined using a BCA assay.

Dried protein extracts (40 µg) were solubilized with 25 µL of 5% SDS. Proteins were submitted to reduction and alkylation of cysteine residues by the addition of tris(2-carboxyethyl)phosphine (TCEP) and chloroacetamide to a final concentration, respectively, of 10 mM and 40 mM. Protein samples were then processed for trypsin digestion on S-trap Microdevices (Protifi) according to the manufacturer's protocol, with the following modifications: precipitation was performed using 211 µL S-Trap buffer; 4 µg trypsin was added per sample for digestion, in 25 µL TEAB 50 mM pH 8.

Tryptic peptides were resuspended in 20 µL of 2% acetonitrile and 0.05% trifluoroacetic acid and analyzed by nano-liquid chromatography (LC) coupled to tandem MS, using an UltiMate 3000 system (NCS-3500RS Nano/Cap System; Thermo Fisher Scientific) coupled to an Orbitrap QExactive Plus mass spectrometer (Thermo Fisher Scientific). Five microliters of each sample were loaded on a C18 precolumn (300 µm inner diameter × 5 mm, Thermo Fisher Scientific) in a solvent made of 2% acetonitrile and 0.05% trifluoroacetic acid, at a flow rate of 20 µL/min. After 5 min of desalting, the precolumn was switched online with the analytical C18 column (75 µm inner diameter × 50 cm, in-house packed with Reprosil C18) equilibrated in 95% solvent A (5% acetonitrile, 0.2% formic acid) and 5% solvent B (80% acetonitrile, 0.2% formic acid). Peptides were eluted using a 5%–50% gradient of solvent B over 115 min at a flow rate of 300 nL/min. The mass spectrometer was operated in data-dependent acquisition mode with the Xcalibur software. MS survey scans were acquired with a resolution of 70,000 and an automatic gain control (AGC) target of 3e6. The 10 most intense ions were selected for fragmentation by high-energy collision-induced dissociation, and the resulting fragments were analyzed at a resolution of 17,500, using an AGC target of 1e5 and a maximum fill time of 50 ms. Dynamic exclusion was used within 30 s to prevent repetitive selection of the same peptide.

Raw MS files were processed with the Mascot software (version 2.7.0) for database search and Proline (89) for label-free quantitative analysis (version 2.1.2). Data were searched against *E. coli* entries of the UniProtKB protein database (release Swiss-Prot 2019_11_05, 23,135 entries). Carbamidomethylation of cysteines was set as a fixed modification, whereas oxidation of methionine was set as a variable modification. The specificity of trypsin/P digestion was set for cleavage after K or R, and two missed trypsin cleavage sites were allowed. The mass tolerance was set to 10 ppm for the precursor and to 20 mmu in tandem MS mode. The minimum peptide length was set to seven amino acids, and identification results were further validated in Proline by the target decoy approach using a reverse database at both a PSM and protein false-discovery rate of 1%. For label-free relative quantification of the proteins across biological replicates and conditions, cross-assignment of peptide ions peaks was enabled inside the group with a match time window of 1 min, after alignment of the runs with a tolerance of ±600 s. The abundance of the Eut proteins in M9 glycerol EA B12 was calculated using percentage intensity based absolute quantification (iBAQ) values (36).

The mass spectrometry proteomics data have been deposited to the ProteomeX-change Consortium via the PRIDE partner repository with the data set identifier PXD048973.

## Metabolomics analysis

### $^1$H-NMR analysis of culture supernatants

*E. coli* K-12 W3110 WT cultures were grown at 37°C under 200 rpm orbital shaking, in 250 mL flasks containing 50 mL M9 glycerol NH$_4$Cl or EA. For the labeling experiment, fully labeled ethanolamine ($^{13}$C$_2$-ethanolamine, InnovaChem, France) was employed in the last culturing step for M9 glycerol EA B12. Samples (500 µL) were collected every 60 min. Bacteria were immediately removed by filtration (Minisart 0.2 mm syringe filter, Sartorius, Germany) and the flow-through was kept at −20°C until further analysis. The flow-through was thawed and mixed (180 µL) with 20 µL of an internal standard containing 2.35 g/L deuterated trimethylsilylpropanoic acid (TSP-d4) solubilized in D$_2$O. The $^1$H-NMR analyses were performed using a quantitative zgpr30 sequence with water pre-saturation prior to acquisition on an Avance III 500-MHz spectrometer. The parameters were as follows: 286K, 128K points, 4 dummy scans, 64 scans, interscan delay of 8.98 s. Three biological replicates were included for each medium. For the ammonium quantification experiment, the filtration flow through (690 µL, diluted 4× in MQ H$_2$O) was mixed with 7.3 µL 4 M H$_2$SO$_4$ and 2.7 µL MQ H$_2$O (90). TSP-d4 dissolved in D$_2$O (60 µL) was added to 540 µL of the mixture. A solvent suppression proton NMR sequence (zggpw5) at 286K with 32K points, 64 scans, 4 dummy scans, and an interscan delay of 10 s was then applied. Quantification was achieved through an external NH$_4$Cl standard curve (200 mM, 8 mM, 2 mM, 0.5 mM, and 0.1 mM NH$_4$Cl diluted in M9 glycerol) analyzed with $^1$H-NMR with the same acquisition parameters and the same sample preparation. Quantification of ammonium in solution was achieved by integration of NMR signals of protons (between 6.9 and 7.4 ppm) attached to nitrogen in an acidic medium, according to the different occurring species (i.e., NH$_4$, NH$_3$D, NH$_2$D$_2$) in a mixture of H$_2$O and D$_2$O (NHD$_3$ and ND$_4$ in negligible amounts were not detected).

### Isotopic profiling of proteinogenic amino acids

The same cultures as for NMR analyses were used. Biomass (500 µL) was harvested at an OD$_{600}$ of 1 corresponding to the mid-exponential phase. The biomass sample was immediately mixed with 3.5 mL pre-chilled (−20°C) quenching solution made of methanol-acetonitrile-H$_2$O (4:4:2) containing 125 mM formic acid. The mixture was incubated at −20°C for at least 2 h and then centrifugated at 4,500 × *g* 4°C for 5 min. The pellet was stored at −20°C until further usage. After thawing, the pellet was evaporated to dryness with an Orbitrap apparatus (Thermo Fisher Scientific, USA) and hydrolyzed for 16 h at 105°C with 500 µL HCl 6 N. Samples were washed twice before being resuspended into 250 µL ultrapure water and diluted (1:700) for the mass spectrometry analysis. LC-MS was performed on an Ultimate 3000 HPLC system (Dionex, CA, USA) coupled to an LTQ Orbitrap Velos mass spectrometer (Thermo Fisher Scientific, USA). Full scan high-resolution mass spectrometry (HRMS) analyses were performed in positive Fourier tansform mass spectrometry (FTMS) mode, the acquisition parameters and data analysis pipeline being identical to that previously described by Heuillet et al. (91). IsoCor (92) allowed correcting for natural isotopic abundances to determine the carbon isotopologue distributions.

## Modeling

### Extracellular uptake and production fluxes

Glycerol and ethanolamine uptake fluxes, acetate, and ethanol production fluxes, and growth rates were calculated from glycerol, ethanolamine, acetate, ethanol, and biomass concentration–time profiles using PhysioFit (v1.0.1 [41], https://github.com/

[MetaSys-LISBP/PhysioFit](MetaSys-LISBP/PhysioFit)). Ethanol evaporation was considered when calculating ethanol production flux, as detailed in Peiro et al. (93) using an evaporation constant of 0.0379 h$^{-1}$ that was determined experimentally.

### Genome-scale modeling of glycerol and ethanolamine co-metabolism

The iML1515 *E. coli* genome-scale model (43) was first updated by incorporating the EA assimilation reactions catalyzed by Eut enzymes (EutBC, EutG, EutE, EutD as well as EutQ/P) and the Eut BMC subcellular compartment. We assumed that EutBC, EutG, EutE, and EutD were all encapsulated within the Eut BMC core as suggested previously (20), while EutQ/P were assumed to be cytosolic. We also assumed cofactors [i.e., NAD(H) and CoA-SH] to not exchange between the Eut BMC and the cytosol, thus being recycled internally within the Eut BMC core. We finally assumed that acetate and ethanol originated from the Eut BMCs. Flux balance analyses and flux variability analyses were carried out using *cobrapy* (94) after constraining exchange fluxes with experimental uptake fluxes of *E. coli* K-12 W3110 WT (see Results). The final model iDJ1518 and the scripts used to perform the calculations can be found at [https://github.com/MetaSys-LISBP/ethanolamine_metabolism](https://github.com/MetaSys-LISBP/ethanolamine_metabolism). The genome-scale model is also available from the BioModels database (95) ([http://www.ebi.ac.uk/biomodels](http://www.ebi.ac.uk/biomodels)) with the identifier MODEL2403010003.

### $^{13}$C-metabolic flux analysis

To quantify intracellular fluxes during the growth of *E. coli* on glycerol and ethanolamine, we constructed a dynamic $^{13}$C-flux model following the approach detailed before (96). The model contains 21 reactions, 20 species, and 3 compartments (the environment, the cytoplasm, and the eut BMCs), and represents five processes: (i) growth, (ii) glycerol uptake and conversion into acetyl-phosphate (AcP) by glycolysis, (iii) ethanolamine assimilation and conversion through the eut BMCs, (iv) AcP utilization by the TCA cycle, and (v) acetate, acetaldehyde, ethanol, and ammonia excretion (Fig. 5).

The differential equations, which balance the concentrations of extracellular compounds (biomass, glycerol, ethanolamine, acetate, and ethanol) and intracellular compounds [AcP, AcCoA, ethanolamine, ethanol, acetate, glycerol, NAD(H), and ammonia], were completed with isotopic equations for parameter estimation. As detailed before (97), we considered all reactions (except biomass synthesis) separately for unlabeled and labeled reactants. Fluxes were assumed to be constant over time since the cells were assumed to be at metabolic steady-state during the exponential growth phase. We also took ethanol evaporation into account, which was modeled with a mass action rate law, using the evaporation constant determined experimentally in this study. The final model has 13 free parameters in total. These parameters (*p*) were estimated by fitting to the experimentally determined concentration dynamics of biomass, glycerol, ethanolamine, ethanol, and unlabeled and labeled acetate, by minimizing the objective function f defined as the weighted sum of squared errors: $f(p) = \sum_i \left( \frac{x_i - y_i(p)}{\sigma_i} \right)^2$, where $x_i$ is the experimental value of data point *i*, with an experimental standard deviation $\sigma_i$, and $y_i(p)$ is the corresponding simulated value. The objective function *f* was minimized using the particle swarm optimization algorithm (2,000 iterations with a swarm size of 50). The experimental and fitted data of one biological replicate are shown in Fig. 5, and detailed results for all replicates are provided in Fig. S8; Data S1.

The model was constructed and analyzed using COPASI (98) (v4.27) and is provided in SBML and COPASI formats in Data S1 and at [https://github.com/MetaSys-LISBP/ethanolamine_metabolism](https://github.com/MetaSys-LISBP/ethanolamine_metabolism). The model has also been deposited in the Biomodels database ([https://www.ebi.ac.uk/biomodels](https://www.ebi.ac.uk/biomodels)) (95) with the identifier MODEL2403010002 to ensure reproducibility and reusability. Extensive details on the model are provided in Data S1.

## *Calculation of the flux through individual BMCs*

All the detailed calculations are given in Data S2.

## ACKNOWLEDGMENTS

We thank Isabelle Fourquaux (CMEAB) and Sylvain Cantaloube (CBI-LITC) for their technical help.

This work was funded by grants from INRAE (NANOBEs and ColiMATTERS projects) and by the ANR (FUNCEMM, ANR-23-CE44-0038). We acknowledge the METi imaging facility, a member of the National Infrastructure France-BioImaging supported by the French National Research Agency (ANR-10-INBS-04). The work was also funded in part by grants from the Région Occitanie, European funds (Fonds Européens de Développement Régional, FEDER), Toulouse Métropole, and the French Ministry of Research with the Investissement d'Avenir Infrastructures Nationales en Biologie et Santé program (ProFI, Proteomics French Infrastructure project, ANR-10-INBS-08).

## AUTHOR AFFILIATIONS

[1]Toulouse Biotechnology Institute, Université de Toulouse, CNRS, INRAE, INSA, Toulouse, France

[2]Plateforme de Microscopie Electronique Intégrative, Centre de Biologie Intégrative, Université de Toulouse, CNRS, Toulouse, France

[3]Institut de Pharmacologie et de Biologie Structurale (IPBS), Université de Toulouse, CNRS, Université Toulouse III—Paul Sabatier (UT3), Toulouse, France

[4]Infrastructure nationale de protéomique, ProFI, Toulouse, France

[5]MetaToul-MetaboHUB, National infrastructure of metabolomics and fluxomics, Toulouse, France

## AUTHOR ORCIDs

Denis Jallet http://orcid.org/0000-0003-0201-5671
Odile Burlet-Schiltz http://orcid.org/0000-0002-3606-2356
Pierre Millard http://orcid.org/0000-0002-8136-9963
Stéphanie Heux http://orcid.org/0000-0003-1312-3002

## FUNDING

| Funder | Grant(s) | Author(s) |
|---|---|---|
| Institut National de Recherche pour l'Agriculture, l'Alimentation et l'Environnement (INRAE) | NANOBES, ColiMATTERS | Denis Jallet |
| Agence Nationale de la Recherche (ANR) | ANR-23-CE44-0038 | Stéphanie Heux |
| Agence Nationale de la Recherche (ANR) | ANR-10-INBS-08 | Alexandre Stella |
| | | Odile Burlet-Schiltz |
| Agence Nationale de la Recherche (ANR) | ANR-10-INBS-04 | Stéphanie Balor |

## DATA AVAILABILITY

The generated proteomics data are available from the ProteomeXchange Consortium via the PRIDE partner repository with the data set identifier PXD048973. The iDJ1518 genome-scale model and associated scripts can be found at https://github.com/MetaSys-LISBP/ethanolamine_metabolism. iDJ1518 has also been deposited on the BioModels database (http://www.ebi.ac.uk/biomodels) with the identifier MODEL2403010003. The isotopic model implemented to fit the 13C2-EA labeling data is described extensively in Data S1 and available at https://github.com/MetaSys-LISBP/

ethanolamine_metabolism. The model has also been deposited in the BioModels database with the identifier MODEL2403010002.

## ADDITIONAL FILES

The following material is available online.

### Supplemental Material

**Data S1 (mSystems00750-24-s0001.pdf).** Documentation for the isotopic model of ethanolamine and glycerol co-metabolism in *E. coli*.
**Data S2 (mSystems00750-24-s0002.docx).** Details of the flux calculation through individual BMCs.
**Supplemental figures (mSystems00750-24-s0003.pdf).** Fig. S1-S9.
**Legend (mSystems00750-24-s0004.docx).** Legend for Movie S1.
**Supplemental tables (mSystems00750-24-s0005.docx).** Tables S1 to S5.
**Movie S1 (mSystems00750-24-s0006.avi).** Visualization of the moving Eut BMCs *in vivo* in *E. coli* K-12 W3110.

### Open Peer Review

**PEER REVIEW HISTORY (review-history.pdf).** An accounting of the reviewer comments and feedback.

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
