## [Reviewer comments · mSystems]

Integrative *in vivo* analysis of the ethanolamine utilization bacterial microcompartment in *Escherichia coli*.

Denis Jallet, Vanessa Soldan, Ramteen Shayan, Alexandre Stella, Nour Ismail, Rania Zenati, Edern Cahoreau, Odile Burette-Schiltz, Stéphanie Balor, Pierre Millard, and Stéphanie Heux

Corresponding Author(s): Denis Jallet, Toulouse Biotechnology Institute

Review Timeline:

Submission Date:

June 6, 2024

Accepted:

June 12, 2024

Editor: Ákos Kovács

Reviewer(s): The reviewers have opted to remain anonymous.

Transaction Report:

DOI: <https://doi.org/10.1128/msystems.00750-24>

Re: mSystems00750-24 (Integrative *in vivo* analysis of the ethanolamine utilization bacterial microcompartment in *Escherichia coli*.)

Dear Dr. Denis Jallet:

Your manuscript has been accepted, and I am forwarding it to the ASM production staff for publication. Your paper will first be checked to make sure all elements meet the technical requirements. ASM staff will contact you if anything needs to be revised before copyediting and production can begin. Otherwise, you will be notified when your proofs are ready to be viewed.

Sincerely,
Ákos Kovács
Editor
mSystems